# Distributed Leadership: School Principals' Practices to Promote Teachers' Professional Development for School Improvement

**Marisol Galdames-Calderón** 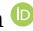

Department of Applied Pedagogy, Autonomous University of Barcelona, 08193 Barcelona, Spain; marisol.galdames@uab.cat

**Abstract:** Distributed leadership is based on increasing the knowledge and skills of those who play the role of leaders at schools. The objectives of this study are (a) to analyze school principals' practices aimed at creating professional development opportunities to promote teacher leadership for school improvement and (b) to relate the perceptions of teacher leaders regarding the professional development opportunities granted by principals. The methods were based on an examination of 21 interviews, including individual semi-structured interviews with school principals and group interviews with teacher leaders at six public schools in Chile, as well as a documentary analysis of institutional educational projects using thematic analysis and NVivo 12 software. The results are presented in three categories: management of principals regarding school organization, development of the professional capacities of teacher leaders, and management of principals regarding school coexistence and the participation of teacher leaders. The discussion and conclusions detailing school principals' practices show that distributed leadership helps develop teachers' leadership competencies. Furthermore, this study suggests that training amplifies the beneficial effects of distributed leadership on teachers' development. Finally, the findings imply that school principals should regularly fulfill their responsibilities and pay attention to teachers' professional development to improve their schools.

**Keywords:** distributed leadership; school principals' practices; teacher leadership; teacher professional development

## 1. Introduction

This study investigated a distributed approach to leadership practices to improve school effectiveness from the perspective of principals and teacher leaders in public schools in Chile. Distributed leadership was examined because it can significantly help in the transformation and improvement of schools under the right conditions [1], as it does not require a single person to perform all the essential management functions. Instead, a group can collectively execute such functions [2].

This reinforces the idea that leadership is the result of collaboration between school leaders and followers while considering aspects of their situation, including tools and routines, from a distributed perspective [3,4] and the theory of social practices [5,6]. Leadership is not a quality that only some people have but is a set of skills and practices in a certain situation that many members of a school have or can acquire.

A distributed perspective on leadership would help to decongest school management, boost school autonomy, promote collaborative work, and enhance participation in institutional decision-making because it provides the possibility of having numerous leaders who can cooperate in both formal and informal capacities [7]. For this reason, principals are key because they best know their staff, school context, and organizational culture, and can appropriately manage their resources [8] to execute core organizational functions through leadership practice [4].

One of the leadership practices of school principals is the creation of professional development opportunities for teachers, particularly those considered by their peers [8]. In this study, such teachers are referred to as "teacher leaders" [9,10]. Teachers also play an important role in leadership practice. Teachers often act as mentors and role models for their peers, helping to create a supportive and collaborative learning environment in schools [11]. They may also be responsible for coordinating and supervising their peers' work and setting and meeting goals to improve their school [9].

*1.1. Theoretical Framework*

The Chilean educational context has been constantly evolving in recent years and has been particularly challenging owing to the COVID-19 pandemic. Regulatory changes, such as the laws on the competitiveness of principals and heads of the administrative department of public education [12], education quality and equity [13], quality assurance system [14], school inclusion [15], teacher professional development system [16], and new public education system [17], have encouraged constant transformations in the actions of people who practice leadership, specifically among school principals.

With the promulgation of the General Education Law [18], the role of the school head is faced with the introduction of new requirements, where the responsibility of raising the quality of education in schools is explicitly required, including encouraging teachers' professional development, meeting institutional goals, considering the regulations and rules established in the Institutional Educational Project (IEP), and carrying out pedagogical supervision in classrooms. In addition, the obstacles presented by the Chilean Educational Reform and its slow implementation of local public education services [17], whether due to administrative difficulties, lack of technical capacity, or the confinement measures of COVID-19, are still factors that influence school leadership, will continue transforming the educational scenarios of public schools in Chile until 2029 [19].

Within the framework of the reform, structural changes take place at the state level, giving special emphasis to principals because they are considered the second factor that most impacts student learning after the work carried out by teachers [20–22]. Principals are expected to have an impact on teaching work through a series of practices, such as the construction and implementation of a strategic vision, facilitation of professional teacher development, effective management of school organizations, encouragement of teacher participation and school coexistence, and pedagogical leadership and training of future teacher leaders [23].

In this sense, the work and bureaucratic burden has increase for school principals, which could cause complications in organizational responses to constant changes in the context and may foster institutional fragility, apathy, and disunity between the different levels and members in an educational community [24]. Therefore, it is necessary to focus on school leadership using a shared approach that allows for greater collaboration, innovation, and creativity within a school, as advocated by the distributed perspective [3,4].

Distributed leadership is an emerging concept in education. This perspective involves sharing the responsibility and authority for school improvement among different people within a school. This model has been gaining traction in recent years, with school districts worldwide [25–29] seeking different methods to train teachers and other members of the school community to participate in the decision-making process.

This is an influential idea in the political and practical fields of education because it is a resistant concept adaptable to different situations for two reasons. First, it considers leadership as a practice, the core idea of which is that leadership occurs through practice rather than through the characteristics or actions of specific individuals [3,4]. Second, it emphasizes the interactions of people (people who take on leadership and those who follow) based on a particular situation instead of being limited to those who have formal leadership positions or roles [7,20,21]. Therefore, in each situation, distributed leadership arises from interaction with others (leaders and followers).

Social practice theory must be used to understand the relationship between situations and leadership practices. According to Ariztía [5], the theory of social practices posits that social practices are collective actions characterized by repetitive engagement. A practice can be defined as a collective, repetitive, and socially situated form of activity embedded in a specific cultural and social context [6]. Social practices are rooted in social relationships and entail an interplay of three constitutive elements: competence, meaning, and materiality. These elements are indispensable for the establishment and functioning of a practice.

The following explanation illustrates this concept using a specific example pertaining to the practice of distributed leadership. Consider an organization that embraces distributed leadership, wherein the responsibilities of leadership are distributed among multiple members rather than concentrating on a single individual. Competence, meaning, and materiality play a pivotal role in this context.

As an essential element, competence encompasses the skills, knowledge, and abilities of team members that enable them to assume leadership roles. Each member must demonstrate the requisite competence to undertake responsibilities and make decisions within the framework of distributed leadership [4]. Proficiency in effective communication, decision making, and problem-solving skills is an indispensable component of competence.

The meaning of leadership practices, as another constitutive element, involves a shared understanding and construction of the significance of a practice. This necessitates that team members comprehend and value distributed leadership as an effective approach to managing and leading collective endeavors. This entails the cultivation of a common vision pertaining to a particular practice and the recognition of its importance in achieving organizational objectives, such as school improvement [30] and the effective improvement of student learning outcomes [31].

Materiality, the third constitutive element, pertains to material and physical resources that are indispensable for the execution of a practice, such as tools, routines, infrastructure, time, space, and/or resources offered by a situation [32]. Within the domain of distributed leadership, materiality encompasses collaboration and communication tools, information systems, and other technological resources that facilitate effective coordination between team members.

The absence or alteration of any material element, such as an inefficient communication system or a lack of access to collaboration tools, can significantly impact distributed leadership practices [4]. For instance, transitioning from a real-time communication system to a slower and less efficient one would compromise the capacity for real-time coordination and decision making, thereby exerting a negative influence on the practice.

Additionally, a variety of physical components, including procedures, tools, frequent staff meetings, and scheduling arrangements, can either enable or constrain practices [32]. In attempts to explain leadership practices that focus on the person thought to be practicing them, these characteristics of the circumstances are frequently disregarded [4]. When equipment and other situational elements are present, they are perceived as accoutrements to a practice rather than as the fundamental components that define it [5].

Competence, meaning, and materiality define the execution of a practice that ceases when an element disappears. A practice involves collective knowledge instead of individual knowledge of a subject [7]. With the practice of social knowledge, people cannot be considered solely responsible for the execution of the practice because they are part of its constitution, and materiality allows the explanation of the existence of a practice in addition to being the context or space of representation [32].

Distributed leadership is a social practice that emerges through the interaction of a social process involving a leader, followers, and a specific situation. In this context, the school principal and teacher leaders constitute the individuals involved, whereas the school environment provides materiality. This is directed towards school improvement [27].

Distributed leadership is associated with other practices, such as developing people [20]. One common practice undertaken by school principals is to create opportunities for teacher professional development to foster teacher leadership, as it is closely linked

to building leadership capacity within a school, thereby enabling leadership practice [25]. Figure 1 illustrates the principal leadership practices undertaken by a school principal, incorporating the constitutive elements outlined in the theory of social practices, namely, competence, meaning, and materiality [5,6], while emphasizing social phenomena and the shared influence process inherent to leadership distribution.

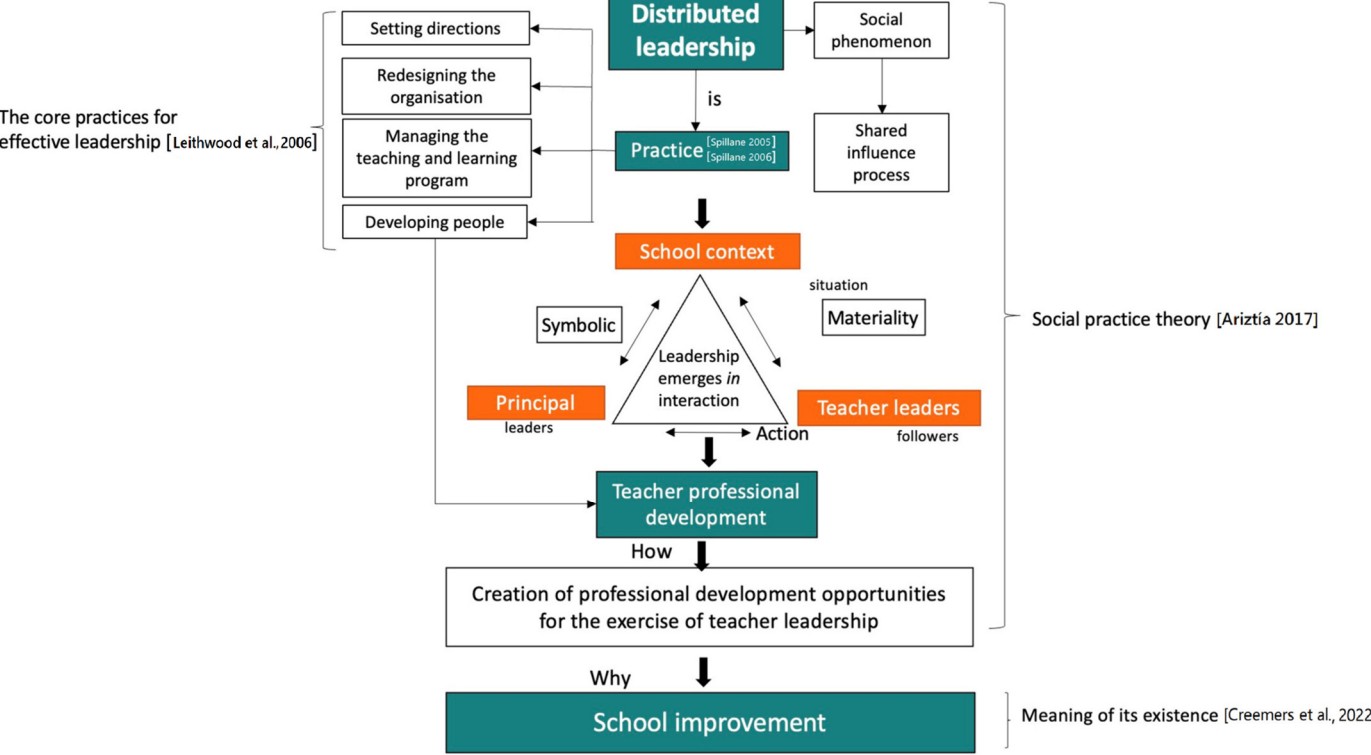

**Figure 1.** Distributed leadership as a social practice. Adapted from Spillane [3,4], Ariztía [5], Leithwood et al. [20] and Creemers et al. [30].

Distributed leadership is based on the idea that all members of an educational community contribute to the success of the school, instead of depending on the central leader [33]. It emphasizes the collaborative and collective nature of leadership, where authority and decision making are dispersed among various individuals within an organization. Unlike heroic leadership [34], which relies heavily on a single leader, distributed leadership acknowledges the diverse expertise and contributions of multiple stakeholders, such as teachers [9,10].

The inclusive and participatory nature of distributed leadership fosters a sense of shared responsibility, empowerment, and ownership among team members [35]. By involving a broader range of perspectives and harnessing a group's collective intelligence, distributed leadership promotes innovative problem-solving, continuous learning, and adaptability to changing educational contexts [33].

Furthermore, distributed leadership aligns with the complex and dynamic nature of the modern educational environment. It recognizes that no single leader possesses all the knowledge and skills required to address the multifaceted challenges in schools. Instead, distributed leadership taps into the expertise and potential of various individuals, allowing for a more comprehensive and responsive approach to school improvement [30].

Moreover, distributed leadership enhances professional development and capacity building within the school community [25]. By encouraging the growth and development of leadership skills among teachers and other stakeholders, distributed leadership nurtures a culture of continuous improvement and expertise. This cultivates a supportive and collaborative environment in which everyone invests in the shared goal of enhancing

student learning outcomes [30,31]. Table 1 presents the key distinctions between traditional hero-focused leadership and distributed leadership.

**Table 1.** Differences between traditional leadership and distributed leadership.

| Traditional Hero-Focused Leadership | Distributed Leadership |
| :---: | :---: |
| Specialized role or position | Shared influence process |
| Individualistic view of leadership | Leadership practice |
| Leadership as supervision of hierarchical control processes | Leadership that arises in interaction processes |
| Concentrating on individual actions | Collaborative and participatory |
| Task delegation | Distribution of responsibilities |
| Hierarchical relationships | Symmetric relationships |
| Responsibility lies with one person: school principal | Responsibility lies with a team: educational community |
| Centralized power | Shared power |

Source: Adapted from Spillane [4], DeFlaminis et al. [25], and Yukl and Gardner [34].

Within this framework, teacher leaders provide valuable insight into the distribution of leadership among school principals. Experienced educators who demonstrate leadership qualities possess a deep understanding of organizational dynamics and can offer first-hand knowledge of how leadership is shared and enacted within the school context. A teacher leader can be defined as an influential educator who not only excels in their classroom, but also takes on additional responsibilities to inspire and support colleagues, contributing to the overall improvement of teaching and learning practices in the school.

In other words, teacher leaders can assume responsibility when they have the opportunity to assume a leadership role and belong to a trusted network of principals and colleagues. They influence their peers without a formal position and, in turn, help principals relate to the rest of the community for school improvement [36].

However, it is necessary to encourage relationships between people in formal and informal positions for the emergence of teaching leadership apart from task delegation as a result of work saturation. Indeed, distribution is different from delegation because it is associated with the transfer of influence through different distribution patterns [3,4,7,20,36,37] in a symmetrical relationship, and is not only associated with the bureaucratic assignment of school administration tasks [38,39].

*1.2. Purpose of the Study*

In spite of the benefits of distributed leadership, there is still a prevalence in educational communities of perceiving leadership from a traditional, individualistic view, thus leaving school principals the only people responsible for the success or failure of the internal processes of their school. Therefore, the questions that guided this study were as follows. How do principals create opportunities to promote professional development and teacher leadership? What do teacher leaders think of these opportunities?

To answer these questions, the following general objectives were stipulated: (a) to analyze school principals' practices aimed at creating professional development opportunities to promote teacher leadership for school improvement and (b) to relate the perceptions of teacher leaders regarding the professional development opportunities.

## 2. Methods

This study examined school principals' practices associated with the creation of professional development opportunities for teachers to take on leadership in public schools in Chile. For this study, a qualitative approach was chosen because it allows for a deeper understanding of complex and subjective phenomena, such as the context and meaning

of the experiences and perceptions of people linked to the study phenomenon [40]. In addition, a case study approach was selected because it is a specific qualitative research method that focuses on a detailed analysis of a particular case [41], thus allowing for exploration and understanding of complex phenomena in a detailed and in-depth manner through inductive reasoning. Another reason is that it is intrinsic because it provides a detailed and rich description of a situation that arouses interest. A case study approach is also particularistic, that is, it focuses on a particular phenomenon, as well as heuristic, as it allows flexibility and adaptation to new findings during the investigation [42,43].

*2.1. Participants*

The sample focused on principals and teacher leaders at six public educational institutions in the municipality of Colina de Santiago, Chile. The selection of principals and schools was determined using the method of access possibility, considering factors such as willingness to participate, geographical proximity, and availability of resources. This approach ensures feasible and practical access to gather data for the study within the given constraints and resources [42]. In addition, principals selected teacher leaders through snowball sampling, as they were the persons who contacted and recruited teachers [44]. Teachers identified as informal leaders by their school principals were selected for this study. Teachers holding formal leadership positions were excluded from the study.

The participating school principals voluntarily chose to participate in the study because of their intrinsic interests in distributed leadership. Their motivation to engage in research was driven by their recognition of the significance and potential benefits of distributed leadership in educational contexts. As this study employed a case study approach, the representation of the sample solely reflects individuals who willingly participate and do not claim to represent the entire population of school principals [42,43].

The schools that participated in the study were diverse, encompassing primary, high, and vocational education and training (VET) schools. Schools belonging to the public system were chosen because they need the most support to reinforce educational quality, as they welcome students from the most vulnerable sectors [19,20].

According to the Chilean Ministry of Education [45], primary schools in Chile are educational institutions that provide education to students from kindergarten to eighth grades. Primary schools typically have diverse curricula that incorporate both academic and non-academic activities to foster the holistic development of young students. Schools that encompass both primary and secondary education, commonly referred to as "high schools", offer a comprehensive educational experience. These schools cater to students in the ninth to twelfth grades and prepare them for further academic pursuit and vocational pathways. High schools provide a more specialized curriculum, enabling students to select subjects aligned with their interests and future career goals [45].

Vocational education and training (VET) schools in Chile focus on providing specialized vocational education to students who wish to acquire practical skills and competencies for specific occupations. These schools offer a range of vocational programs in areas such as automotive technology, construction, culinary arts, healthcare, and information technology [45]. Table 2 presents the characteristics of the participating schools, including their typology, number of enrolled students, number of contracted teachers, and number of teacher leaders.

**Table 2.** Characteristics of participating schools and participants.

| School | Typology | School Size by Number of Students Enrolled | School Size by Number of Teachers | Number of Teacher Leaders |
|---|---|---|---|---|
| School A | Primary school | 300 | 30 | 2 |
| School B | Primary school | 300 | 30 | 3 |
| School C | Primary school | 800 | 30 | 3 |
| School D | Primary school | 1000 | 30 | 2 |
| School E | High school | 1000 | 30 | 2 |
| School F | Vocational education and training high school | 1100 | 40 | 3 |
| Total: 21 participants | | 6 school principals | | 15 teacher leaders |

## 2.2. Procedure

Individual and group semi-structured interviews and documentary analysis were conducted. The semi-structured interviews were used to collect information that helped to obtain a deep and detailed understanding of the phenomenon. Specifically, because of its flexibility and depth, and the possibility of interaction that the conversation offers, a semi-structured interview enables the opportunity to ask new questions and obtain more open and honest answers [44].

The planning and application of individual and group semi-structured interviews began with the identification of two study variables: the principals' practices associated with the creation of professional development opportunities for teacher leadership for school improvement and the teacher leaders' perceptions of those opportunities. Three categories were then defined: the principals' management regarding school organization, development of the professional capacities of teacher leaders, and the principals' management regarding the coexistence and participation of teacher leaders. Subsequently, the identified dimensions, operational elements, and indicators enabled the creation of questions for interviewing the principals and teacher leaders. The interviews were conducted separately with principals and teacher leaders, with guidelines containing the same thematic areas to convey the principals' practices and perceptions of the teacher leaders.

The documentary analysis allowed the collection of data from written records that were both publicly available and official records of the State Administration, such as the Framework for Good Management and School Leadership (FGMSL) [46] and the Institutional Educational Project (IEP). The first document is a general policy issued by the Ministry of Education that provides guidelines exclusively for school principals on how to fulfil their roles. The IEP consisted of internal documents specific to each school, which were typically created by members of the school community. Such documents are often publicly accessible to public schools. They were chosen because they provide guidance on the main practices of people involved in school improvement processes through leadership and because they provide information on the objectives pursued by the participating schools and how the institutions are structured.

## 2.3. Ethical Approval

The initial contact to gain access to fieldwork began with a direct approach, with members representing the education sector of the Municipal Corporation of Colina requesting permission to invite the school principals to participate. The principals who volunteered to participate were contacted, and this approach was maintained until the interviews were conducted.

## 2.4. Field Work and Data Analysis

The interviews were conducted in the workspaces of principals and teacher leaders to facilitate collaboration among the participants so that the same context served as input for the description of their experiences and perceptions of distributed leadership practices. To avoid biases inherent to human interaction and to discourage the expression of irrelevant personal facts or opinions, question guidelines and protocols were created to ensure validity and reliability. Analytical guidelines were created for the selected documents.

Analytical procedures based on thematic analysis [47,48] were performed using NVivo 12 software to guide the simultaneous process of data collection, coding, and analysis [49]. Methodologically, the first step began with the collection of information through individual semi-structured interviews with principals, and group interviews with teacher leaders. Audio recordings, made with prior authorization and consent from the interviewees, were transcribed. The transcribed texts and documents, such as FGMSL and IEPs, were saved as a PDF format for use in the software.

The second step involved a repeated reading of the transcribed texts and documents to identify the main ideas and keywords, and to select the units of meaning. The third step involved coding and generating codes for the selected units. The fourth step was

categorization by performing axial coding and grouping the definitive codes into categories. Finally, the categories were refined by gathering a series of related categories and forming larger categories to achieve topic saturation.

### 2.5. Evaluating Trusworthiness of the Study

In this case study, trustworthiness was ensured by adhering to the multiple criteria of credibility, dependability, confirmability, and transferability [50–52]. Meticulous steps were performed to ensure credibility. The interviews were carefully transcribed to avoid limitations, such as selectivity, partiality, bias, and incompleteness. Saturation of the emerging themes was sought by analyzing a sufficient number of interviews and documents.

Current and updated documents were used as data sources to ensure alignment with the research context. NVivo 12 software was used to minimize human errors in coding and categorization. The transcripts were made available for confirmation. Throughout the coding and categorization processes, efforts were made to preserve the richness of words and their connotations, avoiding the imposition of meaning from the researcher's perspective. The interviewees' perceptions and perspectives were prioritized. In doing so, the researcher strived to maintain the authenticity and integrity of participants' voices. These precautions were taken to enhance the dependability and confirmability, promote rigorous analysis and interpretation of the data, and bolster the trustworthiness of the study's findings. Transferability was addressed by providing a detailed description of the study context and participants, which allowed readers to assess the transferability of the findings to similar settings.

### 3. Results

The results are presented in narrative form by themes and their respective relationships with the initial categories, grouped into three broad categories: principals' management of school organization, development of the professional capacities of teacher leaders, and principals' management of school coexistence and the participation of teacher leaders.

### 3.1. Principals' Management Regarding School Organization

This category is linked to the first objective. The subcategories were the roles of teacher leaders, organizational conditions, creating opportunities, and school improvement. In relation to the elements that make up leadership, all the principals agree that some qualities that are revealed in shared actions with their peers and help identify a teacher leader are "initiative, creativity, proactivity, curiosity to create projects, motivation, collaborative capacity, desire to do things, responsibility, and self-confidence". Likewise, they consider the concept of leadership practice challenging to define in rigid terms because it depends on the situation, organizational conditions, and opportunity, wherein the initiative and confidence to assume responsibility are key.

Similarly, in addition to teachers' personal commitment, it is important to consider organizational conditions, which are situational elements that enable leadership practices, such as space, time, resources, and routines. The principals acknowledge that they are willing to use regulations that allow the use of non-teaching hours to ensure the time and space that teachers need, considering that "the times are relative due to the positions they hold as teachers" (BDifem, principal of School B).

For their part, the teacher leaders perceive that their principals are willing to restructure the school to encourage their participation, considering the situational factors that make it possible or not, and the emergence of the practice of leadership because "the activities can be changed from one day to the next if there is no time or money" (FDL3mas, teacher leader at School F).

However, given the complexity associated with managing school demands, it is necessary to create professional development opportunities because leadership distribution is only possible if people are willing to assume responsibility. The principals (ADifem, BDifem, CDifem, DDifem, EDifem, and FDimas) affirmed that they created opportunities

for teachers to continue training either on their own initiative or as a need for school. In response to this, the teacher leaders unanimously responded that they had professional development opportunities and that it was a personal choice if they wanted to take these opportunities or whether they wanted to decide between becoming formal leaders or staying in an informal leadership position. The latter refers to the option of being teachers with leadership in the informal sphere and continuing to maintain this role "behind the scenes" (CDL2fem, teacher leader of School C).

Finally, both principals and teacher leaders agree that leadership is a co-effect of school improvement because the emergence of teacher leadership has a single meaning in its existence: the improvement of student learning. The concept of a co-effect suggests that achieving school improvement requires leadership practice, whereas leadership practice itself requires a clear goal: school improvement. This highlights the interdependence between distributed leadership and teachers' professional development in fostering school improvement. For instance, FDL2mas perceives School F as a place where leadership practices are embraced and combined with targeted professional growth opportunities. This promotes a culture of continuous learning and leads to improved student outcomes and school success (FDL2mas, teacher leader from School F).

For this, it is necessary to establish a base of people with knowledge and a level of specialization that allows them to face constant changes and improvement processes that arise in the school environment because the intention to improve the school is for students (BDL1fem, teacher leader from School B).

The documents analyzed are in line with the ideas described by principals and teacher leaders. The IEPs of the schools dedicate specific sections to express the intentions of improvement, both of the school and of a plan that seeks quality through school change and the relationship that leadership has with the management of the school because "it is an integrated institutional and educational process that, over time, produces an increase in the level of quality in its processes and results in the institution" (FPEI, institutional educational project of School F).

### 3.2. Development of the Professional Capacities of Teacher Leaders

Within this category, we sought to relate the subcategories that allowed us to achieve the second objective of the study: to analyze how principals support teacher professional development for the promotion of teacher leadership. Four subcategories were identified: managerial roles, managerial support, professional teacher development, and promotion of teacher leadership.

It is important to highlight the direct relationship between the subcategories of the principal's role "and support". Both principals and teacher leaders agree that teachers are more likely to manifest leadership if they have the support of their principal (ADifem, School A principal). For example, FDimas explains that "if (a teacher) makes a mistake or crosses the line a little, we support it in a certain way, but we also go back to prosecute where we have to go or how far we can go" (FDimas, Principal of School F). The teacher leaders affirmed that they felt supported by their principal and that there was trust in them when they took initiative.

Regarding the promotion of teacher leadership, there is a consensus among principals and teacher leaders that teachers often adopt a "camouflaged" leadership approach (CDL1fem and CDL3fem). This refers to teachers' preference for exercising informal leadership, rather than formal leadership roles. Teacher leaders prioritize their role as classroom teachers over positions that may place them above their colleagues (CDL2fem). They expressed discomfort with the idea of feeling superior, as it may hinder trust and professional relationships with their peers. Consequently, informal leadership among teachers is seen as a way to maintain trust and camaraderie within the school community, rather than as a means to elevate informal leadership to a formal managerial position.

Indeed, some principals (BDifem and DDifem) mentioned that there are competent teachers to hold formal positions, such as future head teachers, but these teachers like to

remain classroom teachers. In the words of CDL1fem, informal or camouflaged leadership indicates that they act as teacher leaders, assuming responsibilities in different tasks from their role without necessarily having a formal leadership position, that the principal trusts that everything will be "perfect", and that it depends on their personal disposition, regardless of whether they assume promotion to a formal position (CDL1fem, teacher leader C).

*3.3. Principals' Management Regarding School Coexistence and Participation of Teacher Leaders*

In this category, the data related to the teachers' assessments in terms of trust and promotion of collaborative work, which allowed the emergence of teacher leadership, were combined. The following subcategories were developed: relationships between principals and teacher leaders, the climate of trust, collaborative work, distribution of leadership, and limits of leadership.

Creating a positive emotional environment is essential for examining leadership practices from a distributed perspective. Teacher leaders value the need for a climate of trust in the workplace either at a personal level, such as self-confidence, or at an institutional level, such as a trusting principal. In addition, they agree that the distribution of leadership is manifested through the distribution of responsibilities, and that teamwork is key.

However, teacher leaders themselves ensure that they have limited empowerment, and that their influence is limited to a few areas. First, due to the goals of the school organization, the existence of teacher leadership makes sense only if it is based on complying with the goal of improving academic performance. Second, because of "supervision or monitoring by their principal, even though it is outcome-oriented learning, teacher leaders learn how far they can go" (FDimas, Principal of School F).

Third, the fact that leadership is practiced by people other than the school principal does not diminish the importance of the principal in school leadership. This demonstrates that leadership is often a group effort rather than an individual effort. For this reason, the third limitation of teacher leadership is related to the norms of the legal framework of the national context and the hierarchy of the internal regime of schools, because these norms maintain the idea that the principal is the formal person in charge of an educational community and administration, ignoring group effort.

The fourth limitation is related to situational aspects, as these aspects can either enable or fail to enable the practice of leadership. Therefore, management by a principal is essential for restructuring organizational conditions.

Teachers' willingness to assume leadership is also considered a fifth limitation in the practice of leadership because it depends on each teacher's personal motivation and attitude to assume such responsibility. In fact, some teacher leaders share the idea of retaining informal leadership roles, because they want to continue as classroom teachers. As CDL1fem explains, "We are fond of the classroom more than the administrative; for example, I do not know how to give an example of being in charge of school coexistence, guidance, UTP, or another; they are not within our personal interest" (CDL1fem, teacher leader at School C). However, other teacher leaders wanted to continue to improve to reach managerial positions (ADL2fem, DDL1fem, and FDL1fem).

Finally, the sixth limitation is the relationship with other members of the educational community, specifically their peers, as teacher leaders do not like to feel that others change their perceptions after practicing leadership roles when they do not have formal positions.

From documentary analysis, FGMSL [43] is the most important conceptual reference for school principals in Chile because it provides guidelines that help consolidate their role. As such, it defines the main practices, competencies, and knowledge for the development of school leadership. The use of this framework helps manage the coexistence and participation of educational communities, among other factors.

According to the aforementioned documents and IEPs, the primary responsibility of a school director is to assume the leadership and guidance of the institutional educational project and the processes of educational improvement. Specifically, this entails exercising technical-pedagogical leadership. However, in practice, there is a consensus

among principals and teacher leaders that school principals distribute their leadership among various members, encompassing both those occupying formal positions and those fulfilling informal roles. This incongruity prompts a critical examination of the alignment between legal mandates and day-to-day leadership practices in schools, underscoring the prevalence of distributed leadership models that involve a wider array of stakeholders beyond traditional hierarchical positions.

## 4. Discussion and Conclusions

Distributed leadership is a social practice because it is based on the idea that leadership does not reside in just one person or a small group of people but is distributed among all members of a school. This allows both principals and teacher leaders to contribute and make decisions in a more active and meaningful way, which, in turn, can improve the effectiveness and efficiency of their organizations. This study shows that the five elements constitute distributed leadership. The emergence of leadership must be facilitated and supported, and its potential is realized with the most basic premise: the willingness to create opportunities for the development of teacher leadership capacity, which is the first element shaping school principals' practices of distributed leadership.

Three elements are essential to the distributed leadership perspective because, according to Spillane [3,4], it is people who take on leadership roles and those who interact with their situation that allows distributed leadership to emerge. With respect to the three elements of social practice theory—competence, materiality, and meaning [5]—it is possible to merge these elements such that distributed leadership may prevail in the face of a new scenario that may present a different situation and require decision making.

Principals and teacher leaders must have the competencies necessary to know how to use the available materiality, which grants the organizational conditions to carry out leadership in a course that motivates action towards school improvement. If competencies are linked to people, which are the second and third constitutive elements of distributed leadership, the third element is the materiality offered by the situation. The fifth element is meaning, as shown in Figure 2.

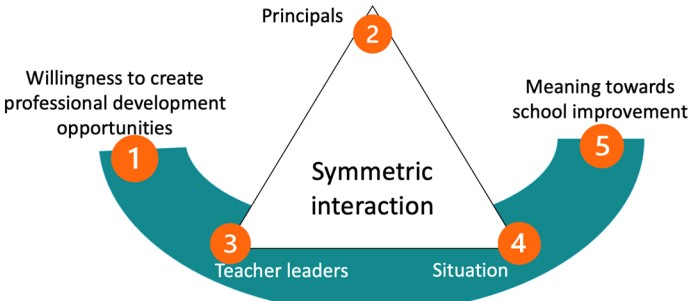

**Figure 2.** Five elements that constitute distributed leadership with teacher leaders at school.

It is important to note that with the redefinition of the role of school principals, management strategies are required to overcome individualistic management through principal support to influence the distribution of leadership. The perceptions of school principals were similar to those of teacher leaders in this study. In fact, teacher leaders agree that support is key when they take initiative and they consider it necessary because it helps develop relationships based on trust and promotes positive professional attitudes, such as openness to feedback and continuous training [33].

This means that the promotion of teacher leadership is beneficial at the personal, professional, and organizational levels because it allows schools to capitalize on experiences and reduces the chances of error, as long as it is acknowledged that empowerment is limited by various factors, such as the goals of the organization, supervision of the principal, regulations, situational aspects, teachers' willingness to assume leadership, and relationships with peers. The practices of school principals associated with the creation of

opportunities for teachers' professional development of leadership practices and linked to school improvement are summarized in Table 3.

**Table 3.** Main practices of school principals.

| Management of Principals Regarding School Organization | Development of the Professional Capacities of Teacher Leaders | Management of Principals Regarding Coexistence and Participation of Teacher Leaders |
| --- | --- | --- |
| Identify teacher leaders | Promote teachers' professional development | Encourage relationships and interactions between principals and teacher leaders |
| Restructure the organizational conditions of the school | Support teacher leaders in professional development | Foster confidence in the school and self-confidence |
| Create opportunities for teachers' professional development | Foster the promotion of teacher leadership | Promote collaborative work |
| Promote a sense of school improvement | Empower teachers as leaders | Promote distributed leadership |

Schools should change their perspective and consider innovative leadership practices. By introducing a new framework and moving away from the traditional leadership perspective, distributed leadership broadens the concept and applications of leadership in schools. Traditional and individualized perspectives have a narrower definition of leadership than the distributed approach, which has the potential to broaden leadership practices, particularly as it resolves the issue of exclusion associated with a traditional vision [3,4,7,8,20,22,24,25,34–36].

In this study, the principals were frank in admitting that it is necessary for teachers to take on leadership roles to manage the school organization effectively and efficiently and that teacher professional development is a decisive factor in the emergence of teacher leadership. In this sense, it is necessary to systematically create opportunities and restructure organizational conditions so that teacher leadership can be successfully implemented.

The teachers' perceptions agreed with the principals regarding the fact that people who participate in the distribution of leadership must have a series of competencies that make them eligible based on their tasks or responsibilities within the school. It is also necessary to have the materiality granted by organizational conditions, such as time, space, and resources, and to recognize that what motivates their activity is the improvement of school effectiveness. In addition, teacher leaders also value that leadership practice can be learned because it is a resource and social knowledge, and in order to accomplish this, it can be acquired either through training courses, recruiting agents outside the school, or empowering members to help the organization via internal training and the distribution of learning among peers.

In summary, adopting a distributed leadership approach presents a transformative opportunity to redefine the leadership roles within schools. By challenging the conventional hierarchical structure dictated by legal requirements, schools can embrace the concept of distributed leadership [36]. Internally, school leaders recognize the need to distribute leadership responsibilities among teachers, providing them with professional development opportunities to enhance their leadership skills and contribute to overall school improvement [33].

This study's implications underscore the importance of further research on distributed leadership. The continued exploration of the benefits, challenges, and effective implementation strategies of distributed leadership practices contribute to the existing knowledge base. Moreover, it is through such research endeavors that we can advocate for policy modifications that recognize and value the collective efforts between school principals and teachers.

**Funding:** This study was funded by MCIN/AEI/ 10.13039/501100011033 and European Union: NextGenerationEU/PRTR. Grant number FJC2021-046504-I.

**Institutional Review Board Statement:** This study adhered to the standards of the Social Sciences of the Ethical Committee of Experimentation of the University of Barcelona (Spain).

**Informed Consent Statement:** Informed consent was obtained from all the subjects involved in the study. All participants were informed that anonymity was assured, why the research was being conducted, and how the data would be used. As with all research involving humans, ethical approval from the University of Barcelona Ethics Committee was obtained before conducting the study.

**Data Availability Statement:** The data presented in this study are available upon request from the corresponding author. The data were not publicly available due to private and ethical restrictions.

**Conflicts of Interest:** The author declares no conflict of interest.

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
