# Peer review of "Distributed Leadership: School Principals’ Practices to Promote Teachers’ Professional Development for School Improvement"

_education, doi:10.3390/educsci13070715_

Round 1

Reviewer 1 Report

I have two overall recommendations and a number of more specific comments about aspects of the paper that are confusing.

Overall Recommendations

My first overall recommendation is to request the authors rewrite the paper(or have it rewritten) in order to improve the English) and resubmit. While the meaning of some language misuse can sometimes be interpreted, much of it cannot.

My second overall recommendation if to simplify the paper’s conceptual framework and the presentation of results related to that framework. The paper has several important - and not especially complicated – objectives and would be much stronger if the results were presented in a more straightforward way.

Detailed Comments

Names of authors are sometimes not cited in the text, for example,  Ref. [5]. In all such cases the citation in the text should include the author’s name, for example, Ariztia [5] 

Page 1 “This invites a change of perspective and considers leadership as a practice and not as an aspect that only some people have, that is, as a social phenomenon in 34 which more members of the school participate”.  This wording seems very awkward to me. I think the author means something like “Leadership is not a quality that only some people have but a set of skills and dispositions many members of a school have or can acquire.”

Page 3, line 14: “the material allows the explanation of the existence of the practices in addition to being the context or space of representation”. I have no idea what this means.

Page 4, Table 1: this table is left unexplained in the text. While some of the table’s contents seem easily understood, others do not as, for example, the difference between “Rationalist-positivist paradigm” and “Descriptive-analytical or normative-prescriptive paradigm”. The text should include a thorough explanation of the table’s contents.

 The sample of schools, principals and teachers summarized in Table 2 needs additional elaboration. Based on the current description, readers cannot determine 

·      how principals and schools were selected 

·      how large are the schools and how representative the selected teachers might be of each school

Addionally, based on the description in the text, noone but the researchers need to know the code used for the principals and teachers. The codes especially in the far right column of the table provide no information of value to the reader. If these codes are intended to provide useful information to the reader, the text should include an explanation of them.

Also, Table 2 lists, as a feature of traditional leadership, that “Responsibility lies with one person: school principal”. This is in contrast to a feature of distributed leadership “Responsibility lies with the team: educational community”. However, line 338 states that “owing to the regulations of the legal framework of the national context and the hierarchy of the internal system of the school, the principal is formally responsible for the educational community and administration”.  I assume this must mean that the results of the study found significant amounts of leadership distribution in the context of a legal framework holding the principal ultimately responsible for whatever happens in the school. So the evidence from the study contradicts this feature of distributed leadership in Table 2.

 Line 278: “ leadership is a co-effect of school improvement because the emergence of teacher leadership has a single meaning in its existence: the improvement of student learning.” Once again, I cannot figure out what this means. And what is a co-effect?

Figure 2: I doubt very much if readers will attempt to decipher this very busy figure. The relationships it captures seem better left to the text.

Line 410:  I never did figure out the meaning of “materiality”.

Table 3 seems straightforward and easy to read. Perhaps this is the core of the results of the study and the more complex figures just confuse matters.

Throughout the paper leadership practice is used in a way that I found confusing. For example, line 432 states: “ To conclude, it is recommended to change perspectives and consider leadership as a practice, not as a psychological aspect that only a few people have”. But leadership is conceptualized early in the paper as embedded in the interactions of people. Much of the leadership literature uses the term “practice” to signify a distinct action or behavior enacted by a person. So is very confusing without some additional clarification

I have no idea what a “psychological aspect” might mean and it is used in the paper in more than a few places.

I have two overall recommendations and a number of more specific comments about aspects of the paper that are confusing.

Overall Recommendations

My first overall recommendation is to request the authors rewrite the paper(or have it rewritten) in order to improve the English) and resubmit. While the meaning of some language misuse can sometimes be interpreted, much of it cannot.

My second overall recommendation if to simplify the paper’s conceptual framework and the presentation of results related to that framework. The paper has several important - and not especially complicated – objectives and would be much stronger if the results were presented in a more straightforward way.

Detailed Comments

Names of authors are sometimes not cited in the text, for example,  Ref. [5]. In all such cases the citation in the text should include the author’s name, for example, Ariztia [5] 

Page 1 “This invites a change of perspective and considers leadership as a practice and not as an aspect that only some people have, that is, as a social phenomenon in 34 which more members of the school participate”.  This wording seems very awkward to me. I think the author means something like “Leadership is not a quality that only some people have but a set of skills and dispositions many members of a school have or can acquire.”

Page 3, line 14: “the material allows the explanation of the existence of the practices in addition to being the context or space of representation”. I have no idea what this means.

Page 4, Table 1: this table is left unexplained in the text. While some of the table’s contents seem easily understood, others do not as, for example, the difference between “Rationalist-positivist paradigm” and “Descriptive-analytical or normative-prescriptive paradigm”. The text should include a thorough explanation of the table’s contents.

 The sample of schools, principals and teachers summarized in Table 2 needs additional elaboration. Based on the current description, readers cannot determine 

·      how principals and schools were selected 

·      how large are the schools and how representative the selected teachers might be of each school

Addionally, based on the description in the text, noone but the researchers need to know the code used for the principals and teachers. The codes especially in the far right column of the table provide no information of value to the reader. If these codes are intended to provide useful information to the reader, the text should include an explanation of them.

Also, Table 2 lists, as a feature of traditional leadership, that “Responsibility lies with one person: school principal”. This is in contrast to a feature of distributed leadership “Responsibility lies with the team: educational community”. However, line 338 states that “owing to the regulations of the legal framework of the national context and the hierarchy of the internal system of the school, the principal is formally responsible for the educational community and administration”.  I assume this must mean that the results of the study found significant amounts of leadership distribution in the context of a legal framework holding the principal ultimately responsible for whatever happens in the school. So the evidence from the study contradicts this feature of distributed leadership in Table 2.

 Line 278: “ leadership is a co-effect of school improvement because the emergence of teacher leadership has a single meaning in its existence: the improvement of student learning.” Once again, I cannot figure out what this means. And what is a co-effect?

Figure 2: I doubt very much if readers will attempt to decipher this very busy figure. The relationships it captures seem better left to the text.

Line 410:  I never did figure out the meaning of “materiality”.

Table 3 seems straightforward and easy to read. Perhaps this is the core of the results of the study and the more complex figures just confuse matters.

Throughout the paper leadership practice is used in a way that I found confusing. For example, line 432 states: “ To conclude, it is recommended to change perspectives and consider leadership as a practice, not as a psychological aspect that only a few people have”. But leadership is conceptualized early in the paper as embedded in the interactions of people. Much of the leadership literature uses the term “practice” to signify a distinct action or behavior enacted by a person. So is very confusing without some additional clarification

I have no idea what a “psychological aspect” might mean and it is used in the paper in more than a few places.

Author Response

Hello,

I hope this message finds you well. 

I just wanted to express my heartfelt gratitude for your observations and recommendations. Your input has been truly invaluable to me.

I wholeheartedly agree with your suggestions and have already made the necessary changes based on your feedback. This article holds a special significance for me as it is my first piece of writing in English at such an advanced level. In addition to implementing your advice, I have also taken steps to improve my English proficiency by availing the services of language experts for editing.

Once again, thank you so much for taking the time to review my work and provide such constructive criticism. Your support and guidance are greatly appreciated. Rest assured, I will continue to work diligently towards further improvement.

If you have any additional suggestions or pointers, please do not hesitate to share them with me. Your expertise is invaluable, and I am eager to learn from you.

Thank you once again, and warmest regards.

Reviewer 2 Report

While this article on the practice of distributed leadership in Chilean schools has potential, it requires substantial revision particularly in terms of its expression of written English.

Specific suggestions for revision:

References are needed for the information in table 1.

Additional references are requires in lines 76-81.

An explanation of the school principals' codes in table 2 and also clairty regarding the typology for schools  A to D is needed (are they all primary schools?)

Clarification of what the documents referred to - it was unclear whether these were specific to the schools or general policy documents.

Reference to grounded theory should be removed. This study does not involve theory building but is based on notions of distributed literature. Thematic analysis appears to have been used so this should be referred to and referenced.

It is not appropriate to refer to validity and reliability in a qualitative study. Instead notions of trustworthiness should be explored including confirmabilty, credibility, dependability and transferability (see Creswell & Clark for example).

The inclusion of a quote from a doctoral study in the results section ( lines 251-253 and line 390) was unusual and should be explained. Was the source related to this research? If so this should be explained.

Both figures 1 and 2 need more explanation including a clearer link to the study results.

The English in this article is not of sufficient quality to make it clearly understood. Many sentences are difficult to follow and some terminology is confusing (the use of the term training for example, which usually refers to initial teacher preparation programmes). Also the frequent use of the term management when I think leadership was being referred to. Paragraphs began with "This ..." and "However, ..." rather than being clear in their focus. A thorough independent edit of the language is needed.

Author Response

(The authors gave the same response as above.)

Round 2

Reviewer 1 Report

The description of the school sample needs clarification. Line 235 states that the sample was voluntary whereas line 257 says that the sample is intentional. Can't be both.

Author Response

Good morning,

Thank you very much for your suggestion. I have already made the proposed change.

Reviewer 2 Report

The article has been improved by the revisions made, however many of the comments made in the previous review on the quality of English language have not been addressed.

See previous review.

Author Response

Dear Sir/Madam,

I hope you are well. I am writing to say thank you in advance for your suggestions.

Regarding the English edition, I have used the service that the MDPI publisher itself offers. I attach the certificate.

I hope that this is enough to be at the expected level.

All the best.
